# Quality Study and Numerical Simulation Analysis of Solid–Liquid Two-Phase Magnetic Fluid Polishing Seven-Order Variable-Diameter Pipe

**DOI:** 10.3390/mi13040500

**Published:** 2022-03-23

**Authors:** Jing Guo, Lin Gui, Wei Hou, Liwei Sun, Yang Liu, Junye Li

**Affiliations:** 1Key Laboratory of Vehicle Transmission, China North Vehicle Research Institute, Beijing 100072, China; jjxiong@yeah.net (J.G.); luckyagui@vip.sina.com (L.G.); houweimh321@163.com (W.H.); 2Ministry of Education Key Laboratory for Cross-Scale Micro and Nano Manufacturing, Changchun University of Science and Technology, Changchun 130022, China; slwcust@163.com (L.S.); lyangcust@163.com (Y.L.)

**Keywords:** variable-diameter pipe, magnetic fluid, numerical simulation, cross experiment, precision polishing

## Abstract

Variable-diameter pipe parts have been widely used in the automobile industry, aerospace industry, and other fields. To enhance the surface quality of variable-diameter pipe and explore the effect of solid–liquid two-phase magnetic fluid (SLTPMF) on polishing, in this paper, the seven-order variable-diameter pipe with symmetrical structure is taken as the research object to carry out experimental research and numerical simulation. The experimental research shows that the best surface roughness is reduced by an order of magnitude to Ra 0.054 μm. The solid–liquid two-phase magnetic fluid polishing (SLTPMFP) technology has reliability and superiority in improving the roughness of variable-diameter pipe parts. The simulation results show that the wall shear stress of solid–liquid two-phase magnetic fluid on the wall surface of the workpiece affects the improvement of roughness. The greater the wall shear force, the better the surface roughness can be obtained. The velocity and dynamic pressure cloud diagram show that the velocity and dynamic pressure on the center axis of the workpiece first increase and then decrease with the flow of the magnetic fluid. The velocity and dynamic pressure on the near wall surface are reduced due to the shear collision with the workpiece. This work can provide technical and theoretical support for the actual production of SLTPMF precision polishing.

## 1. Introduction

With the development of transportation, medical equipment, aerospace technology, automobile manufacturing, and other industries, the precision requirements of mechanical parts become more and more strict. For some complex channels and small diameter on the surface of the parts, such as high pressure common rail parts of aerospace engine, medical equipment in the drainage pipe, nozzle on the engine, energy absorbing components of spacecraft, etc., it is difficult for the traditional precision machining methods to meet the accuracy requirements of the project.

As a kind of non-traditional surface finishing technology, abrasive flow machining (AFM) has attracted more and more attention due to its unique processing method and good processing effect [1,2,3,4]. Guo and other scholars used AFM selective laser melting to analyze and experimentally study the inner surface quality of Inconel 718, which effectively verified the feasibility of AFM to improve the inner surface quality [5]. Nikhil and other scholars studied the inner surface properties of AISI 304 steel pipe using an electrophoretic deposition-assisted polishing method and obtained the best polishing performance and the best reflectivity performance [6]. Yuan and other scholars used the spiral rotating abrasive flow polishing technology to precisely polish the inner surface of the 6061 aluminum alloy cylinder, and the research results showed that the technology significantly improved the polishing efficiency and uniformity [7]. Radkevich and other scholars used the forced electrolytic plasma polishing (FEPP) method to machine the inner surface of the tubular workpiece and obtained a uniform surface [8]. Tyagi and other scholars have studied electropolishing and chemical polishing methods to reduce the surface roughness of the inner surface. The study found that chemical polishing is suitable for any complex shape of additive manufacturing workpiece, and the electropolishing method can effectively reduce the surface roughness of the inner or outer surface degree [9]. Qu and other scholars designed a new type of ultra-precision polisher suitable for the finishing of inner holes of elbows. It has been verified that the polishing instrument can complete the ultra-precise polishing of the inner surface of the elbow to achieve the purpose of reducing the inner surface roughness [10]. Furumoto and other scholars proposed a method of using free abrasive particles to finish the inner surface of the cooling channel in the injection mold, which effectively improved the surface roughness of the inner surface of the cooling channel [11]. Barman and others designed a new polishing tool and finished flat titanium workpieces and found that the new polishing tool can finish workpieces in the nanometer range [12]. Song and others designed a novel magnetorheological (MR) polishing device under a compound magnetic field to machine the inner surface of a titanium alloy tube. The research results show that the polishing performance is excellent at high magnetic field strength, high rotational speed, and high abrasive particle concentration [13]. Peng and other scholars proposed a new type of magnetorheological polishing process for polishing the inner surface of titanium alloy tubes. The research results show that the magnetorheological polishing process can effectively improve surface precision and polishing efficiency [14]. Zhang and others have developed a new magnetic polishing tool for deterministic polishing of inner surfaces. The experimental results confirmed good repeatability and local polishing ability [15]. Grover and other scholars have designed and developed a new magnetorheological honing process for nano-finishing cylindrical inner surfaces with the help of permanent magnets. Studies have shown that the inner surface roughness of cylindrical ferromagnetic workpieces decreased from an initial value of Ra 360 nm to Ra 90 nm after 100 min of finishing [16]. Barman and others used magnetic field-assisted finishing to polish titanium alloy implants, which resulted in better surface quality and wear performance, and prolonged implant life [17]. Huang and other scholars used the EDEM-FLUENT coupling method to calculate the gas-phase transient flow field and the motion, dynamics and collision properties of sand bodies. Using the Archard wear model, the wear amount of the sandblasting machine was calculated and the wear law was analyzed, which provided a theoretical basis for optimizing the design and operation method of the sandblasting machine [18].

Compared with the traditional AFM technology, magnetic particles and external magnetic field are added in magnetic fluid AFM. The magnetic fluid has low viscosity Newtonian fluid under zero magnetic field, but under the action of magnetic field, the fluid shows the characteristics of Bingham body with high viscosity and low fluidity. The magnetic particles in the abrasive under the action of magnetic field can make the abrasive particles contact with the surface to be processed more fully, which makes the polishing efficiency higher. For the properties and machining effects of magnetic fluids, some scholars had given some research conclusions. Kim and other scholars used finite element simulation analysis to study the thermodynamic behavior of magnetic fluids with nano-sized magnetic particles in thin channels. The numerical analysis showed that the magnetic volume force in the thin channel increased with the increase of the magnetic field strength and the viscosity of the magnetic fluid [19]. Parameswari and other scholars discussed the key quality properties of nanoscale polishing on disk by magnetorheological polishing process. The response surface model was established by sequential experimental design. In order to have a uniform surface roughness across the entire surface, the same template was used to measure the surface roughness at the same points before and after finishing. The percentage change of surface roughness was affected by the grinding particle concentration and initial surface roughness, and the regional roughness of Ra 49 nm was realized in the study [20]. Singh and other scholars highlighted a new technique for manufacturing magnetic abrasive materials by mechanical alloying and heat treatment. The properties of these magnetic abrasives were analyzed from the perspective of surface finish improvement of aluminum tubes. The surface finishing experiment of aluminum tube was carried out by response surface method. The results showed that the optimum surface finish of Ra 0.27 μm was obtained [21]. Khan and other scholars explored the issues related to the ball-end magnetorheological finishing and developed fluid compositions suitable for copper finishing. The nano-surface was obtained with very few shallow scratches through experiments [22,23]. Murata and others developed a high-efficiency magnetic-field-assisted polishing process (MFAPP), and polishing using core-shell particles with a magnetic field has shown potential to obtain better material surface quality compared to conventional polishing [24]. Zhang and others have proposed a new magnetically driven polishing tool for inner surface finishing. This new technique uses abrasives driven by spherical magnets and external bar magnets for material removal inside the tube. The research results show that the surface roughness of ring polishing is improved by more than 98%, and the surface roughness of cross-section polishing is improved by 86% [25]. Guo and other scholars proposed a novel rotary vibrating magnetic abrasive polishing method to complete the precision machining of a complex inner surface, revealing the surface evolution mechanism under different motions, improving the material removal efficiency, and reducing the surface roughness degree [26]. Wang and others proposed a novel magnetic field-assisted mass polishing (MAMP) technique for the efficient finishing of multiple free-form components simultaneously. Experimental results show that MAMP is effective for polishing many free-form surfaces with nanometer surface finish [27]. Guo and others used a MFAPP to study RSA 905 aluminum alloys, which resulted in high material removal rates and low surface roughness. At the same time, the MFAPP also helps to relieve surface residual stress and improve friction properties [28]. Wang and others used the finishing properties of the magnetic field-assisted mechanochemical polishing process to achieve the internal finishing of Si_3_N_4_ thin ceramic tubes [29].

It can be seen that current scholars have a profound understanding of abrasive flow precision machining technology and have also conducted good research on magnetorheological polishing technology. However, the research on SLTPMF precision polishing is not sufficient, and its application range has not been extended in a wide range. The SLTPMF machining inside the parts with different internal channels has different machining effects, and this paper takes the symmetrical seven-order variable-diameter pipe as the research object to carry out experimental study and numerical simulation. Based on AFM technology and combined with magnetorheological finishing technology, the effects of magnetic field strength, inlet velocity, and abrasive particle size on the precision machining effect of abrasive flow containing magnetic particles are investigated. The research results can provide technical guidance for the future SLTPMF machining of symmetrical variable-diameter pipe parts and contribute wisdom to the development of SLTPMF precision polishing technology.

## 2. Theory and Method 

### 2.1. Kelvin Force and Magnetic Viscosity Coefficient of Ferromagnetic Fluid

In the presence of an external magnetic field, the ferromagnetic fluid will be subjected to Kelvin force during movement.
(1)$$f_{m}=\eta_{r}\chi M\nabla H$$

$f_{m}$ is the magnetic pressure. The ferromagnetic fluid with a volume fraction of about 7% can be approximated as the magnetic susceptibility χ=1. The magnetic force appears in the ferrohydrodynamic equation, such as the ferrohydrodynamic Navier–Stokes equation written in the following form [30,31]:(2)$$\frac{dv}{dt}=-\nabla p+\upsilon \nabla^{2}\nu +\eta_{r}M\nabla H$$
where M is the magnetic magnetization, H is the magnetic field strength, ν is the velocity of the fluid, p is the pressure, υ represents the viscosity of the ferrofluid based-carrier liquid, $\eta_{r}$ represents the suspension viscosity of the magnetic fluid in the presence of an external magnetic field. 

### 2.2. Basic Equations of Ferromagnetic Fluid

In engineering practice, ferromagnetic fluids are usually regarded as incompressible fluids. Here we use the incompressible ferrohydrodynamic equations, which include the ferrohydrodynamic equation derived by Shliomis, the magnetization equation derived from the Fokker–Planck equation, and the Maxwell static magnetic field equation [32]:(3)∇·ν=0
(4)$$\rho \frac{dv}{dt}=-\nabla p+\eta \nabla^{2}\nu +(M\cdot \nabla )H+\frac{1}{2}\nabla \times (M\times H)$$
(5)$$\frac{dM}{dt}=\Omega \times M-\frac{1}{\tau_{B}}\left[M-\frac{3L(\xi )}{\xi }\chi H\right]-\frac{3\chi }{2\tau_{B}M^{2}}\left[1-\frac{3L(\xi )}{\xi }\right]M\times (M\times H)$$
(6)∇·B=0,  ∇×H=0  (B=H+4πM)
where L(ξ) is the Langevin function, ρ is the density of the ferrofluid, η is the viscosity coefficient of the base-carrier liquid, Ω is the vorticity, $\tau_{B}$ is the Brownian magnetization relaxation time, and B is the magnetic flux density.

### 2.3. Method

In this paper, the method of combining numerical simulation analysis and experimental analysis is used to discuss the polishing effect of solid–liquid two-phase magnetic fluid processing on seventh-order reducer.

## 3. Experimental Study on Polishing Seven-Order Variable-Diameter Pipe by SLTPMF

### 3.1. Selection of Workpieces

The seven-order variable-diameter pipe whose aperture decreases to the middle is selected for experimental research. The technical drawing of the seven-order variable-diameter pipe is shown in Figure 1. The material of the selected pipes is 304 stainless steel. The original workpiece is processed by a CNC milling machine, and there are pits, protrusions, and burrs on the surface of the inner hole. The sample diagram of the seven-order variable-diameter pipe is shown in Figure 2.

### 3.2. Experimental Design

This paper studies the precision polishing of seven-order variable-diameter pipe by SLTPMF. Considering the settings of the machine tool, the properties of the abrasive particles and the characteristics of the magnetic particles, the inlet velocity, the magnetic field strength, and the abrasive particle size are selected as the experiment parameters. 

In this study, the SLTPMFP liquid is used for machining. The aviation kerosene containing magnetic particles (Fe_3_O_4_) is used as the continuous phase, and the second phase is SiC particles. According to the research experience of our laboratory, the volume fraction of SiC particles is set to 10% to obtain a better processing effect. Considering the parameters of the machine tool, the inlet velocity during machining is selected as 30 m/s and 50 m/s. The magnetic field applied in this paper is obtained by an electromagnet, which can change the magnetic force of the magnetic field by adjusting the voltage intensity. Because the machined workpiece and the inner flow channel are cylindrical, the magnet is a central hole structure. The force direction of the workpiece under the action of this electromagnet is perpendicular to the wall surface and diverges outwards. The electromagnet used in the research is customized, and the voltage requirement is between 0 and 48 V. The magnetic field strength obtained by adjusting the voltage can be measured by a hand-held magnetometer, and the magnitude of the force on the polishing liquid can be calculated from the measured magnetic field strength. The schematic diagram of the experimental equipment and processing method is shown in Figure 3. In order to make full use of the available range of electromagnet, the magnetic field strength is select when the voltage is 15 V and 45 V. Finally, the existing SiC particles with 500 mesh and 1200 mesh in the laboratory are selected for the experiment. The cross experiment is designed according to the above parameters, and the table of experiment parameters is shown in Table 1.

### 3.3. Surface Morphology Detection and Analysis

According to the previous research experience [33], there is a better machining quality in the middle of the smallest hole area. In this paper, the surface morphology of the smallest hole is selected for observation, and the effect of SLTPMF on the machining effect of the workpiece under different parameters is analyzed. The original sample and four machined samples are observed by stereo microscope. The stereoscopic microscopic of seven-order variable-diameter pipe workpieces are shown in Figure 4.

It can be seen from Figure 3 that the inner surface of the unmachined original 0# is rough, dark, and of low quality. After being polished by SLTPMF, sample #3 has the best brightness, sample #4 is slightly second, and the other two are relatively inferior. This fully shows that by adjusting the experiment parameter combination, the surface quality of seven-order variable-diameter pipe can be improved effectively after polished by SLTPMF. In order to observe the machining morphology more intuitively, the smallest hole is detected by scanning electron microscope. The obtained scanning electron microscope image of the workpieces is shown in Figure 5. 

It can be seen from Figure 5a that the sample not machined by SLTPMF has uneven burrs and varying degrees of damage on the surface. After the initial machining and shaping, the initial horizontal processing traces remain on the surface of the workpiece. After machined by SLTPMF, the surface quality of the workpiece is improved to varying degrees. As shown in Figure 5b,c, the original residual traces on the surface of samples #1 and #2 are eliminated after polishing. However, some rough SLTPMF machining traces are left, which affects the roughness value and surface finish of the workpiece surface. The combination of these two parameters has a certain effect on machining, but it does not improve the surface roughness. For sample #3 and sample #4, the uniform scratches are left on the surface of the workpiece, and the depth of these scratches is not enough to damage the surface quality. There are almost no residual burrs and impurities on the surface of the workpiece, and the surface finish is greatly improved.

### 3.4. Surface Roughness Detection and Analysis

In order to observe the relationship between roughness changes and machining parameters more intuitively, the value of surface roughness Ra is detected by the Mahr stylus surface roughness profiler, and the surface roughness detection diagram is obtained as shown in Figure 6.

In order to visually see the change of roughness, the roughness values detected in Figure 6 are summarized. The roughness data table of the seven-order variable-diameter pipe is shown in Table 2.

The sample #0, which has not been machined by SLTPMF, has a roughness of Ra 0.516 μm. After polishing, the roughness value of sample #3 is the smallest, which is Ra 0.054 μm. Compared with the unpolished workpiece, the surface roughness of the workpieces machined by the SLTPMF is reduced to varying degrees. The best surface roughness is reduced by an order of magnitude. Compared with ordinary abrasive flow precision polishing technology, which has been tested, the SLTPMF polishing technology used in this paper has greatly improved the surface roughness of the workpiece [1,3,33]. The reliability and superiority of SLTPMF machining of variable diameter pipe are proved.

## 4. Numerical Simulation Analysis

The surface roughness of the workpiece obtained by polishing under different parameter combinations is different. In order to theoretically analyze the influence of different machining parameters on the obtained roughness, this paper carries out a numerical simulation of the SLTPMFP seven-order variable-diameter pipe according to the parameters used in the experiment.

### 4.1. Model Building

In this paper, the ANSYS FLUENT software is used to simulate the machining of symmetrical seven-order variable-diameter pipe by SLTPMF. According to the two-dimensional size model of Figure 1, a three-dimensional model diagram of the seven-order variable-diameter pipe is established. The grid is divided, and the quality of the mesh is inspected. The numerical model of the seven-order variable-diameter pipe is shown in Figure 7. 

In the process of numerical simulation, the hexahedral grid is selected to mesh the inner channel region of seven-order variable-diameter pipe. The grid division result of the seven-order variable-diameter pipe inner channel is shown in Figure 7b. As shown in Figure 7c, the quality of the unstructured grid has been tested, and there is no negative volume, indicating that the quality of the grid is reliable. In addition, the mesh quality of the model is greater than 0.3, which meets the accuracy required by the simulation.

### 4.2. Parameter Setting

This article carries out the simulation analysis on the basis of actual experiment conditions. Firstly, the SLTPMF is an incompressible fluid, and the pressure-based solver is selected for the analysis. Choose the Mixture model, in which aviation kerosene containing Fe_3_O_4_ magnetic particles is used as the continuous phase, and the second phase is SiC particles. According to the Reynolds number calculation formula Re=ρvd/μ, the calculated value was less than 2300. Therefore, the laminar flow model is selected for the SLTPMF. For incompressible SLTPMF, the inlet condition is set as the velocity inlet. The speed and pressure of the polishing liquid at the outlet are not easy to measure, so the outlet condition is set as a free outlet. According to actual test conditions, the volume fraction of SiC particles is set to 10%. According to the experiment parameters in Table 1, simulations of four sets of parameters are carried out. 

### 4.3. Simulation Result Analysis

The simulation results in this paper show the shear stress of the SLTPMFP liquid acting on the wall under different parameter conditions. The material removal rate shows a positive linear dependence on shear stress according to the research of Miao and other scholars [34]. The material removal rate also seems to have a positive effect on the machined surface roughness [35,36]. Based on the comparison between wall shear stress and experiment results, this paper gives the comparative relationship between the surface roughness of the workpiece and the wall shear stress after SLTPMF machining. The wall shear stress distribution of the seven-order variable-diameter pipe obtained under the four sets of experiment parameters is shown in Figure 8. In Figure 8, the right side is the inlet, and the left side is the outlet. From left to right are the first to the seventh order.

It can be seen from Figure 8 that the distribution of the wall shear stress cloud diagram under different parameters is basically similar. The wall shear stress in the area with the larger diameter is smaller than that in the area with the smaller diameter. The wall shear stress is the largest in the fourth order, and the value is the smallest in the sixth and seventh orders. Although the diameter of the first three order area is the same as that of the latter three order area, it is obvious that the wall shear stress of the first three order area near the inlet is greater. The fluid in the latter three order area is affected by the smallest aperture from the fourth order, and the wall shear stress is small. Therefore, the area with the best removal effect should be the fourth order with smallest hole area. In the fourth order, the wall shear stress reaches the maximum, the polishing liquid has a better shear effect on the wall, and the bulge and burr on the surface of the workpiece can be removed better.

The change of experiment parameters has little effect on the distribution of wall shear stress, but it greatly affects the numerical value of wall shear stress. As shown in Figure 8c, the wall shear stress is the largest when the inlet velocity is 50 m/s, the magnetic field strength is output when the applied voltage is 45 V and the abrasive particle size is 500 μm. At this time, the material removal rate is the highest. It can also be seen from the scanning electron microscopy image of the experiment results that the smallest hole surface of the sample #3 becomes very smooth after the SLTPMFP, and the roughness value is reduced to the minimum. The order of the maximum wall shear stress of the simulation results is sample #3 > sample #4 > sample #1 > sample #2. Sort the roughness values obtained from the experiment results and obtain sample #3 < sample #4 < sample #1 < sample #2. Therefore, it can be explained that for a seven-order variable-diameter pipe polished by the SLTPMF, the greater value of the wall shear stress can obtain the lower surface roughness value. The machining with large wall shear stress has a positive effect on improving the surface roughness of the workpiece. On the other hand, the large wall shear stress has an impact on the change of the workpiece size, which is also the direction that needs to be studied in future work. In order to further show the flow state of the SLTPMF in the seven-order variable-diameter pipe, this paper obtains the velocity cloud diagram and dynamic pressure cloud diagram of the fluid inside the workpiece during the polishing process. The flow state at 50 m/s, 45 V, and 500 μm is shown in Figure 9.

It can be seen from Figure 9a that the dynamic pressure in the first three order area near the inlet is relatively small, and the largest dynamic pressure is in the fourth-order area. The reduction of the aperture makes the movement of the abrasive more violent, and the dynamic pressure at the central axis increases first and then decreases. After the magnetic fluid flows out from the fourth section area, due to the effect of velocity inertia, the dynamic pressure keeps the maximum at the intersection of the fourth order and fifth order. In the fifth to seventh order zones, the aperture gradually increases to weaken the dynamic pressure, but it is still higher than the dynamic pressure in the first three order area. This shows that the dynamic pressure of the magnetic fluid flowing out through the small hole has been enhanced. It can be seen from Figure 9b that the velocity of the fluid in the direction of the central axis of the pipe is lower in the first three order area near the inlet and reaches the maximum in the fourth-order area. The velocity value of the latter three order areas near the outlet is still relatively large, which is the velocity inertial effect of the polishing liquid after it flows out of the small hole. With the increase of velocity, the energy carried by the fluid is increased and produces a more powerful polishing effect on the wall. Unfortunately, the fluid dynamic pressure and velocity near the wall are relatively small. This is because the SLTPMF will inevitably collide with the wall surface after entering the flow channel. After the impact, most of the fluid kinetic energy is lost near the wall, and then converted into cutting energy. The fluid that is not involved in the actual machining is lost in the direction of the center axis of the flow channel. In terms of the difference between the dynamic pressure and velocity between the center position of each order areas and the position near the wall, the radial value of the fourth order area changes drastically. More fluid kinetic energy is converted into cutting energy, which also shows that the magnetic fluid has the strongest processing effect at the fourth order.

## 5. Conclusions

In this paper, the seven-order variable-diameter pipe with a symmetrical structure is polished by SLTPMF. Through cross experiment and numerical simulation, the following conclusions are obtained:(1)The experimental results show that the optimal surface roughness is reduced by an order of magnitude, reaching Ra 0.054 μm. After polishing, the protrusions and burrs on the surface of the workpiece are eliminated, and the surface of the workpiece becomes smooth. SLTPMFP technology has reliability and superiority in improving the roughness of reducing pipe fittings.(2)The numerical simulation results show that the shear stress of the magnetic fluid on the workpiece under different parameter combinations affects the surface roughness of the workpiece. The greater the shear stress value, the lower the surface roughness of the workpiece. Due to the shear collision between the abrasive particles and the surface close to the wall, the speed and kinetic energy of the abrasive particles are reduced, and the polishing effect is weakened.

## Figures and Tables

**Figure 1 micromachines-13-00500-f001:**
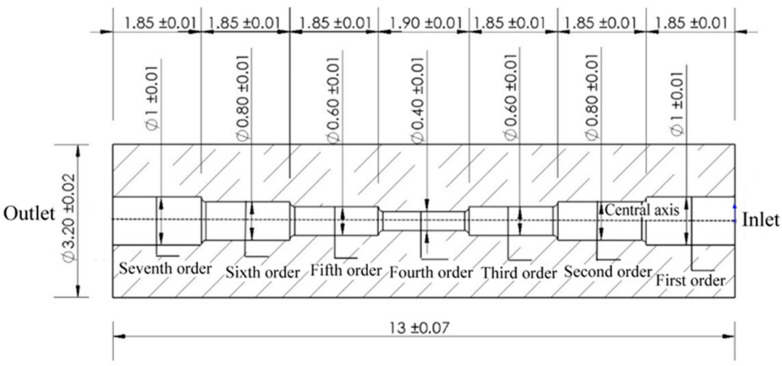
Technical drawing of the seven-order variable-diameter pipe.

**Figure 2 micromachines-13-00500-f002:**
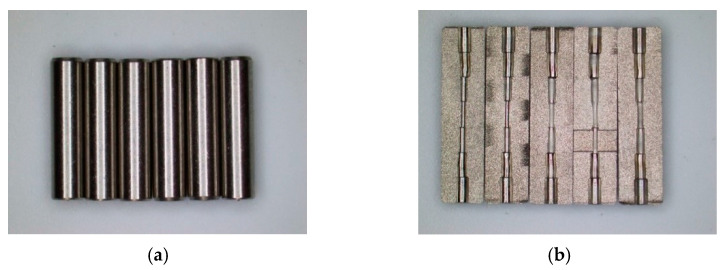
Sample diagram of the seven-order variable-diameter pipe. (**a**) Original sample; (**b**) The sample after wire cutting.

**Figure 3 micromachines-13-00500-f003:**
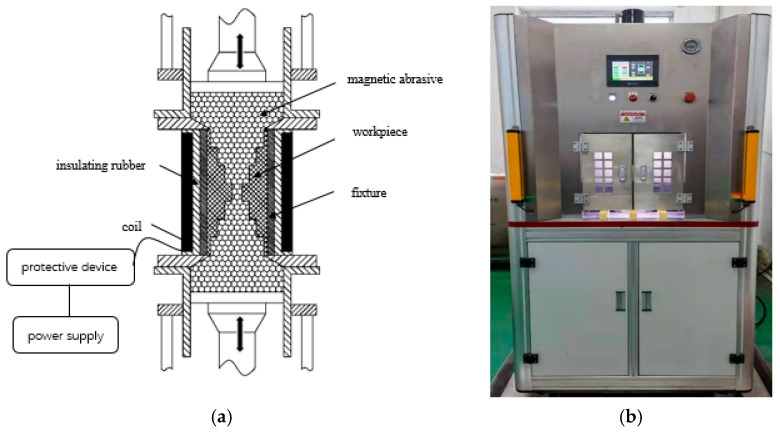
Processing method and experimental equipment diagram. (**a**) Schematic diagram of processing method; (**b**) Laboratory equipment.

**Figure 4 micromachines-13-00500-f004:**
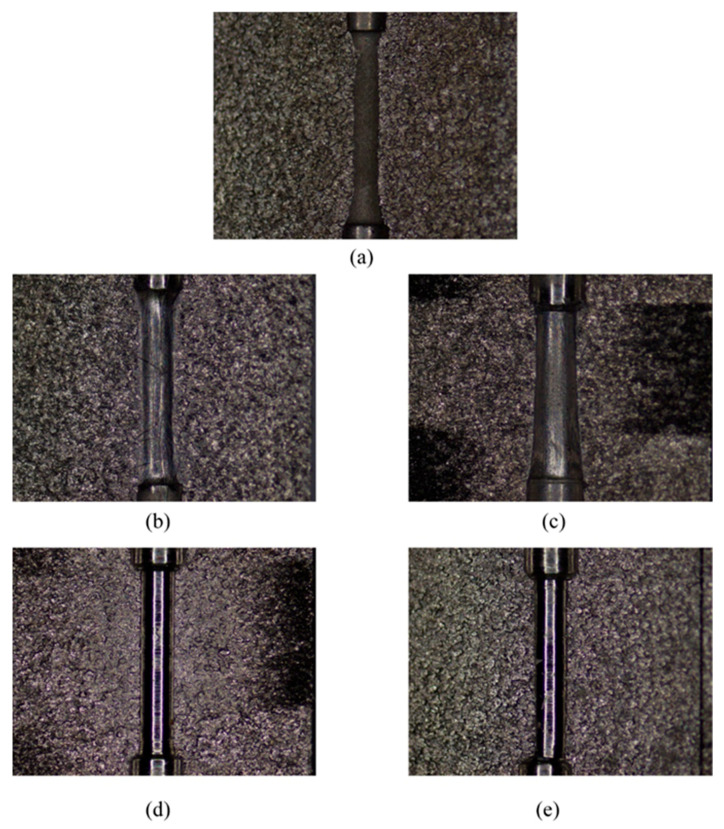
Stereoscopic microscopic of seven-order variable-diameter pipe workpieces. (**a**) Original #0; (**b**) Sample #1; (**c**) Sample #2; (**d**) Sample #3; (**e**) Sample #4.

**Figure 5 micromachines-13-00500-f005:**
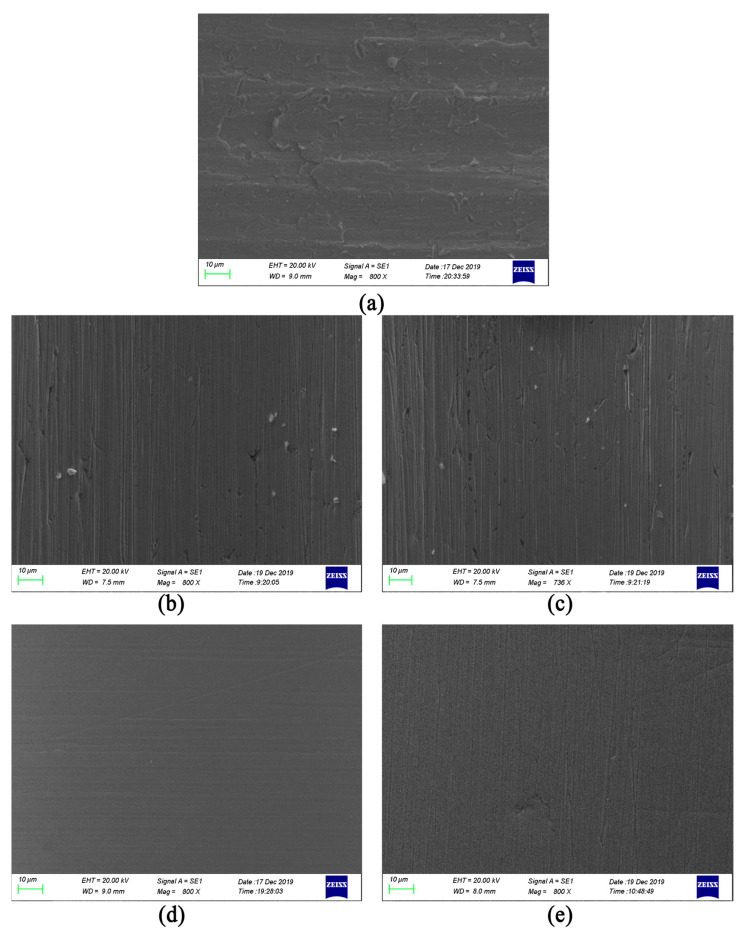
Scanning electron microscope image of the workpieces. (**a**) Original #0; (**b**) Sample #1; (**c**) Sample #2; (**d**) Sample #3; (**e**) Sample #4.

**Figure 6 micromachines-13-00500-f006:**
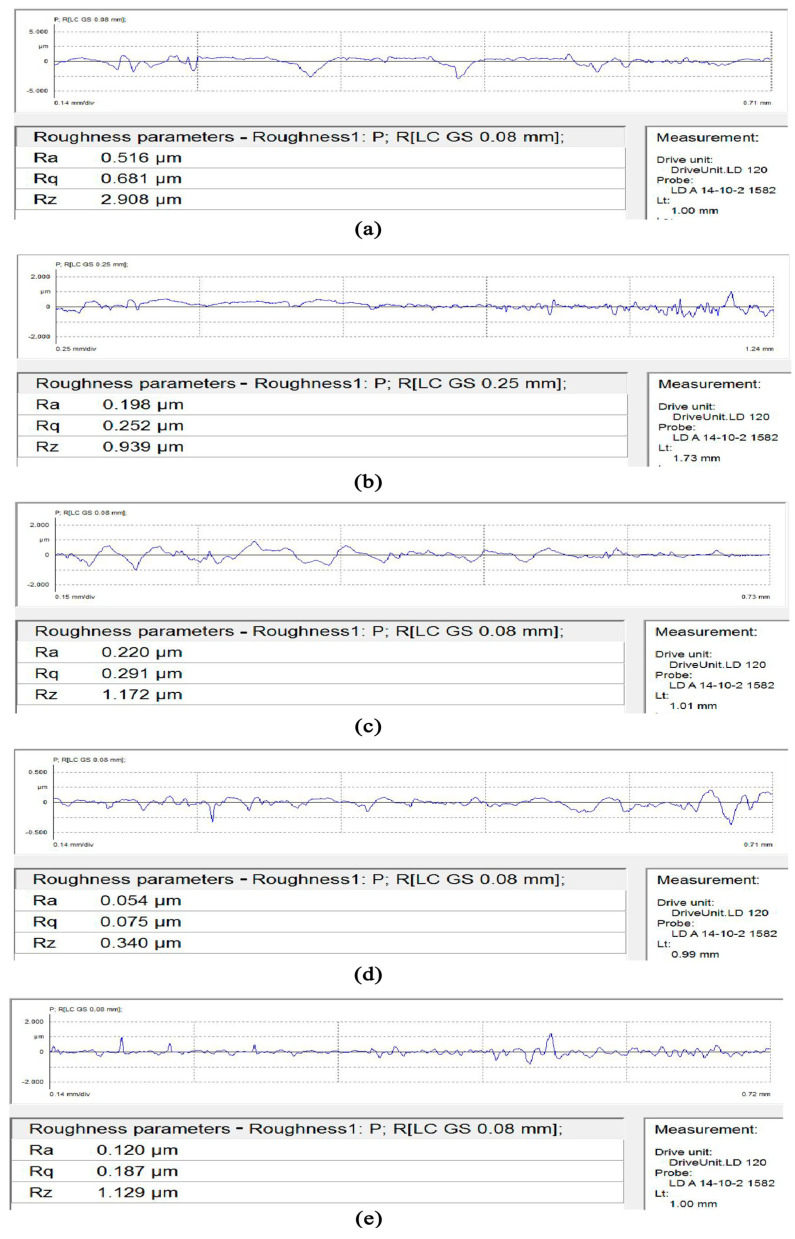
Surface roughness Ra detection diagram. (**a**) Original #0; (**b**) Sample #1; (**c**) Sample #2; (**d**) Sample #3; (**e**) Sample #4.

**Figure 7 micromachines-13-00500-f007:**
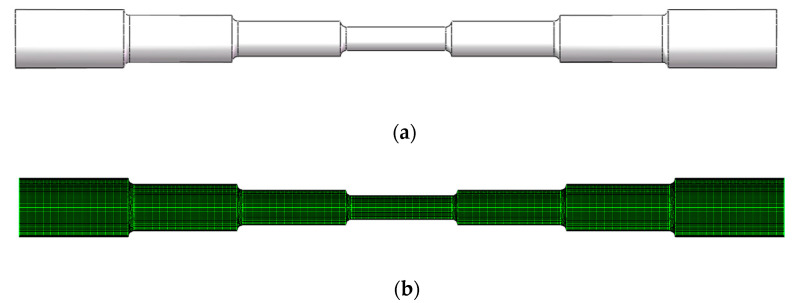
Numerical model of the seven-order variable-diameter pipe. (**a**) Three-dimensional model; (**b**) Grid division; (**c**) Grid quality check.

**Figure 8 micromachines-13-00500-f008:**
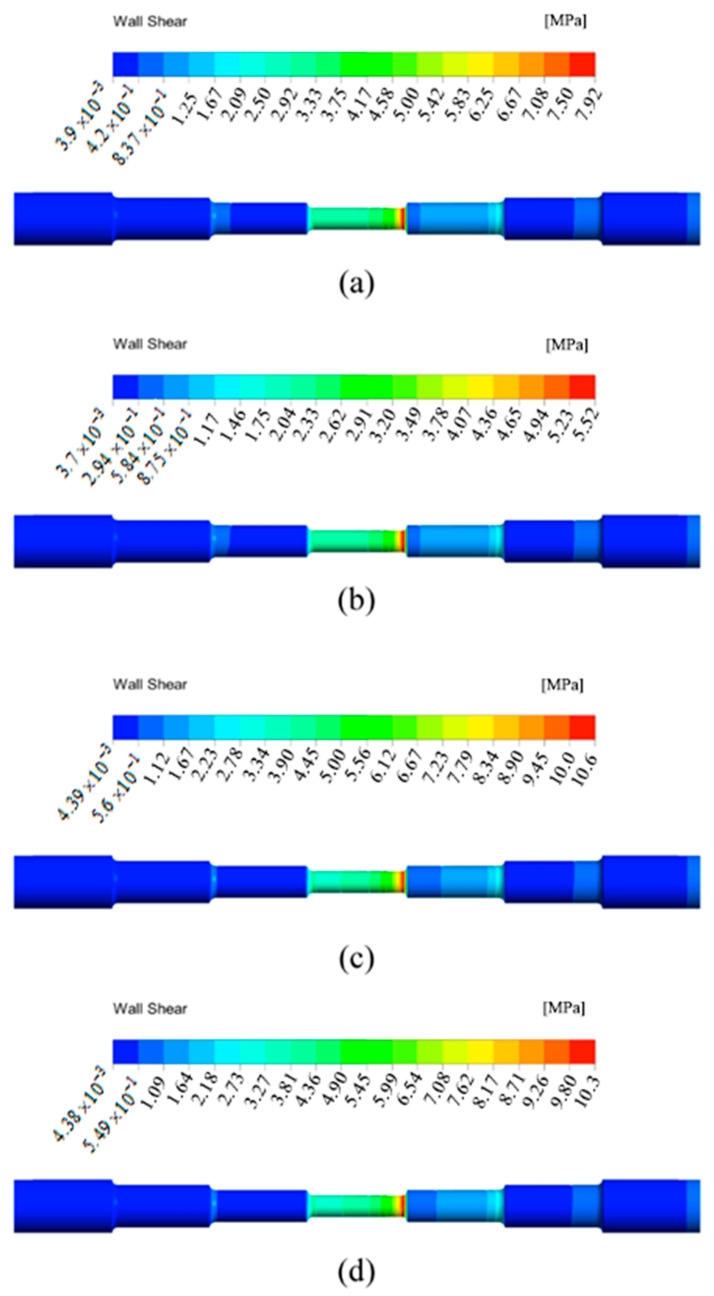
Wall shear stress distribution of the seven-order variable-diameter pipe obtained under the four sets of experiment parameters. (**a**) Under the experiment parameter of 30 m/s, 15 V, and 500 μm; (**b**) Under the experiment parameter of 30 m/s, 45 V, and 500 μm; (**c**) Under the experiment parameter of 50 m/s, 45 V, and 500 μm; (**d**) Under the experiment parameter of 50 m/s, 45 V, and 1200 μm.

**Figure 9 micromachines-13-00500-f009:**
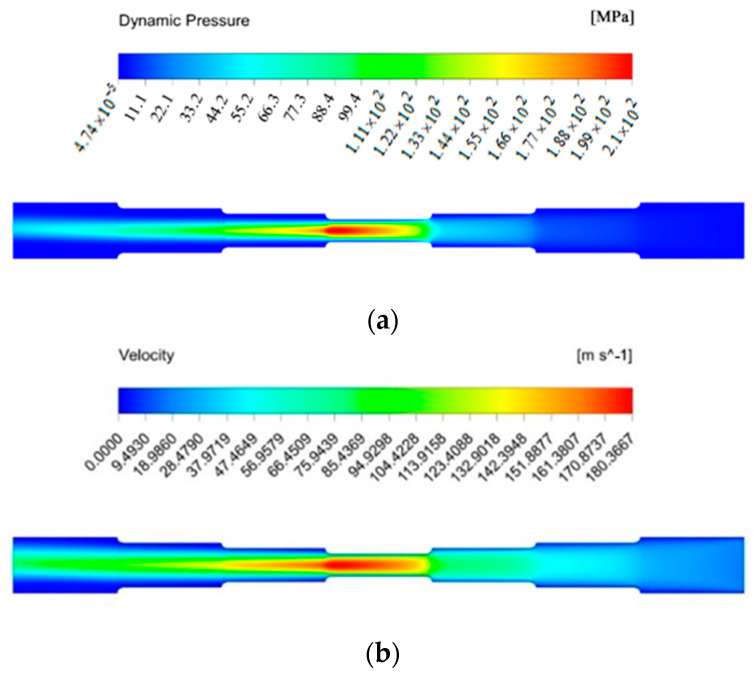
The flow state diagram at 50 m/s, 45 V, and 500 μm. (**a**) dynamic pressure cloud diagram; (**b**) velocity cloud diagram.

**Table 1 micromachines-13-00500-t001:** Experiment parameters table.

Experiment Number	Inlet Velocity (m/s)	Voltage (V)	Abrasive Particle Size (μm)
Sample #1	30	15	500
Sample #2	30	45	500
Sample #3	50	45	500
Sample #4	50	45	1200

**Table 2 micromachines-13-00500-t002:** Surface roughness Ra values.

Experiment Number	Roughness (μm)
Original 0#	0.516
Sample 1#	0.198
Sample 2#	0.220
Sample 3#	0.054
Sample 4#	0.120

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
