# Peer review of "Quality Study and Numerical Simulation Analysis of Solid–Liquid Two-Phase Magnetic Fluid Polishing Seven-Order Variable-Diameter Pipe"

_micromachines, 2022, doi:10.3390/mi13040500_

Round 1

Reviewer 1 Report

1. The keywords are too long, which should be revised.
2. A schematic diagram showing the polishing method in this study is missing, which is hard to understand how the magnetic field is loaded.
3. It is highly recommended to add the photograph of the experimental setup.
4. The quality of Fig.5 should be improved.
5. The literature review can be improved. I believe a brief review on internal/inner surface polishing is needed, since the target surface belongs to this type of surface. And there are many different polishing methods such as magnetic field assisted polishing[1], fluid jet polishing[2], and others. 
[1] Guo J, Au KH, Sun CN, Goh MH, Kum CW, Liu K, Wei J, Suzuki H, Kang R. Novel rotating-vibrating magnetic abrasive polishing method for double-layered internal surface finishing. Journal of Materials Processing Technology. 2019 Feb 1;264:422-37.
[2] Cheung CF, Wang CJ, Cao ZC, Ho LT, Liu MY. Development of a multi-jet polishing process for inner surface finishing. Precision Engineering. 2018 Apr 1;52:112-21.
6. Review on the magnetic field assisted polihsing is far from enough, you can check the following refs:
[1]Wang C, Cheung CF, Ho LT, Yung KL, Kong L. A novel magnetic field-assisted mass polishing of freeform surfaces. Journal of Materials Processing Technology. 2020 May 1;279:116552.
[2]Yamaguchi H, Shinmura T. Internal finishing process for alumina ceramic components by a magnetic field assisted finishing process. Precision Engineering. 2004 Apr 1;28(2):135-42.
etc.

Reviewer 2 Report

The article deals with modified abrasive flow machining. The process was enhanced by the magnetic field. The pipes with different diameters were used as workpieces. The surface quality of the smallest diameter was evaluated. The results were supported by numerical simulation.

The article is written on a high level; however, it contains some mistakes, such as:

- it seems there were not used standard MDPI template (with numbered lines).
- when you mention the value of surface roughness, you should specify the roughness parameter (like Ra). (This appears in abstract, introduction (near [15], near [19]), before Chapter 4, conclusion.)
- you should consider using some acronym for “solid-liquid two-phase magnetic fluid precision polishing”. It appears a lot.
- the explanation of variables for equations should appear after equation 1 as well.
- the variables M and H in Eq.2 have the same explanation. There is a dot on the left side of Eq.2.
- you should mention how cavities were made before the AFM process (in Chapter 3).
- check the use of the word “diagram”. I believe “technical drawing” is more suitable for Fig.1, and “samples” for Fig.2.
- please avoid lines that contain only one word/number (ch.3.1, p.4), as well as labelling figures on another page (Fig.1), and chapter titles in the last line of the page (ch.3.4). You should consider adding additional space before chapter titles.
- it would be better to convert “mesh” into “µm (micrometers)” (use S.I. units only) (page 5, Fig.7, Fig.8).
- units for Magnetic field strength is usually A/m (Amperes per meter), and not V (Volts). Perhaps you should recalculate V to A/m.
- sample number one should be written as Sample #1 (instead of Sample 1#).
- I believe you miss the word “image” in the last sentence on page 6 (SEM image).
- in page 7 there are two almost identical sentences. Perhaps you should consider to combine them into one sentence.
- how many repetitions were performed for roughness measurements? The last sentence above Tab.2 seems to be incomplete.
- the third sentence below Tab.2 is not necessary – it is obvious from Tab.2, where it is more clear. On the other hand, you should quantify the shear stress and dynamic pressure in text, since its values are difficult to read from Fig. 7 & 8a. Or change Pa to MPa in the figures.
- did you analyse surface roughness for other diameters of the pipes?
- why it was necessary to use a 3D model for FEM analyse of the rotational model?
- were the Reynolds numbers calculated for all diameters?
- you used a different text-align for page 11.
- please check the singular for the words “area” and “zone” in the last paragraph before Chapter 5.

Reviewer 3 Report

I have reviewed this interesting manuscript, The authors have conducted a well designed study; and introduced a novel work. The manuscript fits well within the scope of the journal. Hence It will be a very useful addition to the journal; therefore my recommendations are to publish this manuscript after the following minor corrections.

Author should address following comments to improve the contents

Cite references for the 1st paragraph and many other statements in the introduction

Introduction is too detailed, some of the irrelevant text can be deleted.

Section 2 may have the main heading, “methods”

Please highlight limitations of the study.

Conclusion section should be condensed to keep information focused.

Round 2

Reviewer 1 Report

I'm fine with the authors revision. The fontsize inside some figures can be further enlarged.